# Online Convex Optimisation:
# The Optimal Switching Regret for all Segmentations Simultaneously

**Stephen Pasteris**
The Alan Turing Institute
London UK
spasteris@turing.ac.uk

**Chris Hicks**
The Alan Turing Institute
London UK
c.hicks@turing.ac.uk

**Vasilios Mavroudis**
The Alan Turing Institute
London UK
vmavroudis@turing.ac.uk

**Mark Herbster**
University College London
London UK
m.herbster@cs.ucl.ac.uk

## Abstract

We consider the classic problem of online convex optimisation. Whereas the notion of static regret is relevant for stationary problems, the notion of switching regret is more appropriate for non-stationary problems. A switching regret is defined relative to any segmentation of the trial sequence, and is equal to the sum of the static regrets of each segment. In this paper we show that, perhaps surprisingly, we can achieve the asymptotically optimal switching regret on every possible segmentation simultaneously. Our algorithm for doing so is very efficient: having a space and per-trial time complexity that is logarithmic in the time-horizon. Our algorithm also obtains novel bounds on its dynamic regret: being adaptive to variations in the rate of change of the comparator sequence.

## 1 Introduction

We consider the classic problem of online convex optimisation: a problem with numerous real-world applications. In this problem we have an *action* set which is a bounded convex subset of some euclidean space. On each trial we select an action from this set and then receive a convex function of bounded gradient, which associates the action with a *loss*. The aim is to minimise the cumulative loss. The static regret is defined as the cumulative loss of the algorithm minus that of the best constant action in retrospect. It has been shown that the minimax static regret is $\Theta(\sqrt{T})$ where $T$ is the time horizon, and that it is achieved by the classic *mirror descent* family of algorithms [2]. However, in dynamic environments a more sensible notion of regret is the *switching regret*, which is defined relative to any segmentation of the trial sequence and is equal to the sum of the static regrets over all segments. Clearly, if the segmentation is known a-priori then the minimax switching regret is $\Theta(\sum_k \sqrt{\Lambda_k})$ where $\Lambda_k$ is the length of the $k$-th segment, and it is obtained by running mirror descent independently on each segment. *Tracking algorithms*, instead, attempt to bound the switching regret on every possible segmentation of the trial sequence simultaneously. However, as far as we are aware, the best such bound until now was $\mathcal{O}(\sum_k \sqrt{\Lambda_k \ln(T)})$ which is a factor of $\mathcal{O}(\sqrt{\ln(T)})$ higher than the optimal if we knew the segmentation a-priori. In this paper we (quite remarkably) get rid of this factor: hence obtaining the asymptotically optimal switching regret of $\mathcal{O}(\sum_k \sqrt{\Lambda_k})$ for every possible segmentation simultaneously. Not only is our algorithm optimal, but it is also parameter-free and efficient: having both a space and per-trial time complexity of $\mathcal{O}(\ln(T))$.

38th Conference on Neural Information Processing Systems (NeurIPS 2024).

In fact, our algorithm RESET is a meta-algorithm which utilises any *base algorithm* for the online convex optimisation problem at hand. Using *online gradient descent* [11] as our base algorithm gives us the above $\mathcal{O}(\sum_k \sqrt{\Lambda_k})$ switching regret bound. However, the constant under the $\mathcal{O}$ is dependent on the action set and the possible gradients. By choosing a more appropriate base algorithm that is tailored to the specific problem we can achieve lower constant factors. In particular, when faced with the classic problem of *prediction with expert advice* with $N$ experts, using *Hedge* [5] as our base algorithm yields the asymptotically optimal $\mathcal{O}(\sum_k \min(\Lambda_k, \sqrt{\ln(N)\Lambda_k}))$ switching regret (a novel result in itself).

We note that although, like *strongly adaptive* algorithms [4], we are adaptive to heterogeneous segment lengths, we are not necessarily strongly adaptive: in that we do not bound the static regret on any particular segment.

Whilst switching regret models discrete changes in the environment, a continuously changing environment is better modeled by the notion of *dynamic regret*, which is the difference between the loss of the algorithm and that of any comparator sequence of actions. It is known that algorithms exist which bound the dynamic regret by $\mathcal{O}(\sqrt{(1+P)T})$ where $P$ is the *path length* of the comparator sequence. However, this bound is not adaptive to variations in the rate that the comparator sequence is changing. RESET, with online gradient descent as the base algorithm, rectifies this: improving the dynamic regret to $\mathcal{O}(\sum_k \sqrt{(1+P_k)\Lambda_k})$ for any segmentation in which the path length in the $k$-th segment is $P_k$. We note that this implies the $\mathcal{O}(\sum_k \sqrt{\Lambda_k})$ bound on switching regret. However, since we are forced to use online gradient descent as our base algorithm here, this result is not strictly more general than our switching regret result.

**Related works:** *Mirror descent* was introduced in [2] to find minimisers of convex functions in convex sets. The same algorithm, however, can also be applied to online convex optimisation: the *Hedge* algorithm of [5] implementing a special case when the convex set is a simplex and the convex functions are linear (the so-called *experts* problem). The $\mathcal{O}(\sqrt{T})$ static regret of Mirror descent was shown to be optimal in [1]. The work [6] studied the non-stationary case in the experts setting: modifying Hedge to give an algorithm *Fixed share* which takes a parameter $\Phi$ and has a switching regret of $\mathcal{O}(\sqrt{\Phi T \ln(T/\Phi)})$ for any segmentation with $\Phi$ segments. One issue with Fixed share, however, is that it does not adapt to heterogeneous segment lengths. In order to remedy this, [4] gave a strongly adaptive algorithm which achieved a static regret of $\mathcal{O}(\ln(T)\sqrt{\Lambda})$ on any segment of length $\Lambda$. This was improved to $\mathcal{O}(\sqrt{\ln(T)\Lambda})$ in [8]. The work [9] took parameters $a, b \in \mathbb{N}$ and achieved a static regret of $\mathcal{O}(\sqrt{(1+\ln(b/a))\Lambda})$ for any segment of length $\Lambda \in [a, b]$. However, this still leads to a switching regret of $\mathcal{O}(\sum_k \sqrt{\Lambda_k \ln(T)})$ for general segmentations. Our work finally achieves the optimal $\mathcal{O}(\sum_k \sqrt{\Lambda_k})$. The work [11] showed that gradient descent achieves a dynamic regret of $\mathcal{O}((1+P)\sqrt{T})$, which was improved to $\mathcal{O}(\sqrt{(1+P)T})$ in [10]. However, prior to both these works, the work [7] obtained the $\mathcal{O}(\sqrt{(1+P)T})$ bound subject to an optimal parameter tuning. Our work dramatically improves on this bound by being adaptive to variations in the rate of change of the comparator sequence.

**Notation:** Let $\mathbb{N}$ be the set of natural numbers excluding $0$. Given $A \in \mathbb{N}$ we define $[A] := \{a \in \mathbb{N} \,|\, a \leq A\}$, we define $\Delta_A := \{\boldsymbol{a} \in [0,1]^A \,|\, \sum_{i \in [A]} a_i = 1\}$, and we define $A\mathbb{N}$ to be the set of natural numbers that are multiples of $A$. Given a predicate $p$ we define $\llbracket p \rrbracket := 0$ if $p$ is false and define $\llbracket p \rrbracket := 1$ if $p$ is true. Given $q, s \in \mathbb{N}$ with $s \geq q$ let $\langle q, s \rangle := \{a \in \mathbb{N} \,|\, q \leq a \leq s\}$.

## 2 Problem and Results

In this section we introduce the online convex optimisation problem and state the results of this paper. In particular we define and compare the notions of switching and dynamic regret, giving the bounds obtained by our meta-algorithm RESET. Another common notion of regret, not necessarily bounded by RESET, is *strongly adaptive regret* which we discuss in Section 3.3.

### 2.1 Online Convex Optimisation

Here we describe the classic problem of *online convex optimisation*, which our meta-algorithm RESET solves. In this problem we have known bounded convex subsets $\mathcal{X}, \mathcal{G}$ of some euclidean

space. We define $\mathcal{L}$ to be the set of all convex functions that map $\mathcal{X}$ into $\mathbb{R}$ and whose sub-gradients lie in $\mathcal{G}$. The problem proceeds in $T$ trials. On each trial $t \in [T]$ the following happens:

1. We choose some *action* $\boldsymbol{x}_t \in \mathcal{X}$.
2. We receive some *loss function* $\ell_t \in \mathcal{L}$.

Our aim is to minimise the cumulative loss:

$$\sum_{t \in [T]} \ell_t(\boldsymbol{x}_t) \,.$$

Without loss of generality we shall assume that for all $t \in [T]$ and all $\boldsymbol{x} \in \mathcal{X}$ we have $\ell_t(\boldsymbol{x}) \in [0, 1]$. This is without loss of generality as both $\mathcal{X}$ and the sub-gradients of $\ell_t$ are bounded and our algorithm RESET, when using mirror descent as the base algorithm, is invariant to any constant addition to any loss function. Also, without loss of generality, assume that $T$ is an integer power of two.

An example of online convex optimisation is *prediction with expert advice*. Here we have some $N \in \mathbb{N}$ and a set of $N$ *experts*: where on each trial each expert is associated with a loss in $[0, 1]$. On each trial we must select an expert (incurring the loss associated with that expert) and then observe the vector of losses for that trial. For this problem we choose $\mathcal{X} := \Delta_N$ and $\mathcal{G} := [0, 1]^N$. On each trial $t$ we draw our expert from the probability vector $\boldsymbol{x}_t$ and define the loss function $\ell_t$ to be the linear function such that for all $i \in [N]$ we have that $\ell_t(\boldsymbol{e}_i)$ (that is, the loss of the $i$-th basis element of $\mathbb{R}^N$) is the loss associated with expert $i$ on trial $t$. Note that $\ell_t(\boldsymbol{x}_t)$ is our expected loss on trial $t$.

## 2.2 Switching Regret

We now define the notion of *switching regret*. A *segment* is any set of the form $\langle q, s \rangle$ for $q, s \in [T]$ with $s \geq q$. The *static regret* with respect to such a segment $\mathcal{I}$ is defined as:

$$R(\mathcal{I}) := \max_{\boldsymbol{x}^* \in \mathcal{X}} \sum_{t \in \mathcal{I}} (\ell_t(\boldsymbol{x}_t) - \ell_t(\boldsymbol{x}^*))$$

which is the total loss incurred on the segment minus that which would have been obtained by always choosing the best constant action in retrospect. A *segmentation* $\mathcal{S}$ is defined as any partition of $[T]$ into segments. The *switching regret* with respect to such a segmentation $\mathcal{S}$ is defined as:

$$R^\dagger(\mathcal{S}) := \sum_{\mathcal{I} \in \mathcal{S}} R(\mathcal{I})$$

which is the sum of the static regrets on each segment of $\mathcal{S}$. Note that $R^\dagger(\mathcal{S})$ is the total loss of the algorithm minus that which would have been obtained by the best sequence of actions which is constant over each segment of $\mathcal{S}$. The following theorem establishes a lower bound on the switching regret with respect to any fixed segmentation, even in the special case in which all the loss functions are linear:

**Theorem 2.1.** *For any segmentation $\mathcal{S}$ and any algorithm for the online convex optimisation problem, there exists (except in trivial cases) a sequence:*

$$\langle \ell_t \mid t \in [T] \rangle \subseteq \mathcal{L}$$

*of linear functions, in which:*

$$R^\dagger(\mathcal{S}) \in \Omega \left( \sum_{\mathcal{I} \in \mathcal{S}} \sqrt{|\mathcal{I}|} \right)$$

*where the constant under the $\Omega$ is dependent only on $\mathcal{X}$ and $\mathcal{G}$.*

*Proof.* Apply the static regret lower bound of [1] to each segment independently. $\qquad\square$

In this paper we develop an algorithm RESET which has an upper-bound that matches this lower bound for every possible segmentation $\mathcal{S}$ simultaneously. RESET utilises any algorithm (called the *base algorithm*) for the online convex optimisation problem at hand. The base algorithm must take a parameter $\Lambda \in [T]$ and guarantee that $R([\Lambda]) \in \mathcal{O}(\sqrt{\Lambda})$ if it were used directly. We note that online gradient descent [11] is always one such possibility. Computationally, to use online gradient descent, we must be able to compute subgradients of the loss functions and euclidean projections into the set $\mathcal{X}$. The following theorem bounds the switching regret of RESET.

**Theorem 2.2.** *Suppose* $\gamma \in \mathbb{R}$ *is such that for all* $\Lambda \in [T]$, *when the base algorithm is run with parameter* $\Lambda$, *it is guaranteed that:*

$$R([\Lambda]) \leq \gamma\sqrt{\Lambda}\,.$$

*Then, for any segmentation* $\mathcal{S}$, RESET *achieves a switching regret of:*

$$R^\dagger(\mathcal{S}) \leq (c\gamma + d) \sum_{\mathcal{I} \in \mathcal{S}} \sqrt{|\mathcal{I}|}$$

*where:*

$$c := \sqrt{2}/(\sqrt{2}-1) \quad ; \quad d := \sqrt{8\ln(2)}/(3 - 2\sqrt{2})\,.$$

*Proof.* See Section 4. □

Clearly, theorems 2.1 and 2.2 show that, for any fixed pair $(\mathcal{X}, \mathcal{G})$, RESET has the asymptotically optimal switching regret for every segmentation simultaneously. However, our result is stronger. For example, take the problem of prediction with expert advice defined above. Here the HEDGE algorithm attains:

$$R([\Lambda]) \in \mathcal{O}\left(\min\left(\Lambda, \sqrt{\ln(N)\Lambda}\right)\right)$$

which is shown to be optimal via (the proofs of) theorems 3.6 and 3.7 of [3] and by noting that we can always force $\mathcal{O}(\Lambda)$ regret if $\Lambda \leq \ln(N)$. Although there is a slight technicality here when segments have length less than $\ln(N)$, the proof of Theorem 2.2 still works in exactly the same way to show that (by applying RESET to HEDGE) we have the asymptotically optimal switching regret for every value of $N$ and every segmentation simultaneously.

RESET is also very efficient, as shown in the following theorem.

**Theorem 2.3.** *Given that the base algorithm runs in a time of* $\xi$ *per trial and requires a space of* $\xi'$, RESET *has a per-trial time complexity of* $\mathcal{O}(\xi \ln(T))$ *and space complexity of* $\mathcal{O}(\xi' \ln(T))$.

*Proof.* Immediate from the RESET algorithm. □

## 2.3 Dynamic Regret

Switching regret measures the performance of the algorithm against the best comparator sequence of actions that is constant in each segment. Dynamic regret, on the other hand, measures the performance of the algorithm against any comparator sequence. Specifically, given any sequence of actions:

$$\mathcal{E} = \langle \boldsymbol{\epsilon}_t \,|\, t \in [T+1] \rangle \subseteq \mathcal{X}$$

then the *dynamic regret* with respect to $\mathcal{E}$ is defined as:

$$R^*(\mathcal{E}) := \sum_{t \in [T]} (\ell_t(\boldsymbol{x}_t) - \ell_t(\boldsymbol{\epsilon}_t))\,.$$

To bound the dynamic regret of RESET we introduce the following notion of *path length*. Specifically, given the above sequence $\mathcal{E}$ and a segment $\mathcal{I}$, the *path length* of $\mathcal{E}$ in the segment $\mathcal{I}$ is defined as:

$$P(\mathcal{E}, \mathcal{I}) := \sum_{t \in \mathcal{I}} \|\boldsymbol{\epsilon}_{t+1} - \boldsymbol{\epsilon}_t\|_2\,.$$

The current state of the art for dynamic regret is the algorithm ADER [10] which achieves a dynamic regret of:

$$R^*(\mathcal{E}) \in \mathcal{O}\left(\sqrt{(1 + P(\mathcal{E}, [T]))T}\right)\,.$$

In this paper we significantly improve on this result, as shown in the following theorem.

**Theorem 2.4.** *When using online gradient descent [11] as the base algorithm,* RESET *achieves, for any comparator sequence* $\mathcal{E}$ *and any segmentation* $\mathcal{S}$, *a dynamic regret of:*

$$R^*(\mathcal{E}) \in \mathcal{O}\left(\sum_{\mathcal{I} \in \mathcal{S}} \sqrt{(1 + P(\mathcal{E}, \mathcal{I}))|\mathcal{I}|}\right)$$

*where the constant under the* $\mathcal{O}$ *is dependent only on* $\mathcal{X}$ *and* $\mathcal{G}$.

*Proof.* See Section 4 ◻

Note that, for any segmentation, the dynamic regret bound of RESET is asymptotically equal to that of running ADER on each segment independently. To achieve this with ADER one would need to know the specific segmentation a-priori. As for the switching regret, RESET achieves this for every segmentation simultaneously. We note that our bound is a significant improvement on that of ADER since it is adaptive to variation in the rate of change of the comparator sequence.

We note that, given a segmentation $\mathcal{S}$, the switching regret $R^\dagger(\mathcal{S})$ is equal to the maximum dynamic regret $R^*(\mathcal{E})$ across all sequences $\mathcal{E}$ that are constant in each segment of $\mathcal{S}$. For all $\mathcal{I} \in \mathcal{S}$, the fact that such an $\mathcal{E}$ is constant on $\mathcal{I}$ implies that the path length $P(\mathcal{E}, \mathcal{I})$ is in $\mathcal{O}(1)$. This means that Theorem 2.4 implies the switching regret bound of Theorem 2.2 up to a constant factor (dependent on $\gamma$, $\mathcal{X}$ and $\mathcal{G}$). However, to obtain Theorem 2.4 we must use online gradient descent as our base algorithm. For prediction with expert advice, for example, online gradient descent is not asymptotically optimal for every value of $N$ simultaneously. Hence, when considering switching regret only, Theorem 2.2 is a stronger result.

# 3 The Algorithm

In this section we describe our meta-algorithm RESET (**Re**cursion over **Se**gment **T**ree). We first introduce the notation that we will use to describe the base algorithm.

## 3.1 The Base Algorithm

We now define the notation that we use to describe our base algorithm. The base algorithm utilises a data-structure $\mathcal{D}$ (which contains the parameter) and is composed of the following three subrountines:

- Given $\Lambda \in [T]$, the subroutine INITIALISE($\Lambda$) returns the initial data-structure with parameter $\Lambda$.

- At the start of a trial, given the current data-structure $\mathcal{D}$, the subroutine QUERY($\mathcal{D}$) returns the output action of the base algorithm for that trial.

- At the end of a trial $t$, given the current data-structure $\mathcal{D}$ and the loss function $\ell_t$, the subroutine UPDATE($\mathcal{D}, \ell_t$) returns the updated data-structure (ready for the next trial).

The assumption in Theorem 2.2 implies the following. Suppose we have trials $q, s \in [T]$ with $s \geq q$, and on each trial $t \in \langle s, q \rangle$ we do the following:

1. If $t = s$ then $\mathcal{D}_t \leftarrow$ INITIALISE($q - s + 1$).
2. $\boldsymbol{w}_t \leftarrow$ QUERY($\mathcal{D}_t$).
3. $\mathcal{D}_{t+1} \leftarrow$ UPDATE($\mathcal{D}_t, \ell_t$).

Then we have:

$$\max_{\boldsymbol{x}^* \in \mathcal{X}} \sum_{t=q}^{s} (\ell_t(\boldsymbol{w}_t) - \ell_t(\boldsymbol{x}^*)) \leq \gamma \sqrt{q - s + 1} \,. \tag{1}$$

## 3.2 RESET

We now introduce our meta-algorithm RESET. First let $\tau := \log_2(T)$ and define the function $\psi : \mathbb{R} \times \mathbb{R} \times \mathbb{R} \times \mathbb{N} \to \mathbb{R}$ by:

$$\psi(\rho, a, b, \Lambda) := \frac{\rho \exp\left(-a\sqrt{2\ln(2)/\Lambda}\right)}{\rho \exp\left(-a\sqrt{2\ln(2)/\Lambda}\right) + (1 - \rho) \exp\left(-b\sqrt{2\ln(2)/\Lambda}\right)} \,.$$

The pseudocode of RESET is given in Algorithm 1.

We now describe RESET. We have a set of $\tau + 1$ *levels*, where each level $i \in [\tau] \cup \{0\}$ hosts an instance of the base algorithm with parameter $2^i$ and which will reset every $2^i$ trials. On each trial $t$,

---
**Algorithm 1** RESET
---
$\quad$ **for** $i \in [\tau] \cup \{0\}$ **do**
$\qquad \mu_1^i \leftarrow 1/2$
$\qquad \mathcal{D}_1^i \leftarrow \text{INITIALISE}(2^i)$
$\quad$ **end for**
$\quad$ **for** $t \in [T]$ **do**
$\qquad$ **for** $i \in [\tau] \cup \{0\}$ **do**
$\qquad\quad \boldsymbol{w}_t^i \leftarrow \text{QUERY}(\mathcal{D}_t^i)$
$\qquad$ **end for**
$\qquad \boldsymbol{z}_t^0 \leftarrow \boldsymbol{w}_t^0$
$\qquad$ **for** $i \in [\tau]$ **do**
$\qquad\quad \boldsymbol{z}_t^i \leftarrow \mu_t^i \boldsymbol{w}_t^i + (1 - \mu_t^i)\boldsymbol{z}_t^{i-1}$
$\qquad$ **end for**
$\qquad \boldsymbol{x}_t \leftarrow \boldsymbol{z}_t^\tau$
$\qquad$ **for** $i \in [\tau] \cup \{0\}$ **do**
$\qquad\quad$ **if** $t \in 2^i \mathbb{N}$ **then**
$\qquad\qquad \mu_{t+1}^i \leftarrow 1/2$
$\qquad\qquad \mathcal{D}_{t+1}^i \leftarrow \text{INITIALISE}(2^i)$
$\qquad\quad$ **else**
$\qquad\qquad \mu_{t+1}^i \leftarrow \psi(\mu_t^i, \ell_t(\boldsymbol{w}_t^i), \ell_t(\boldsymbol{z}_t^{i-1}), 2^i)$
$\qquad\qquad \mathcal{D}_{t+1}^i \leftarrow \text{UPDATE}(\mathcal{D}_t^i, \ell_t)$
$\qquad\quad$ **end if**
$\qquad$ **end for**
$\quad$ **end for**
---

we denote the data-structure of the base algorithm associated with level $i$ by $\mathcal{D}_t^i$. Each level also has an associated number in $[0, 1]$ called the *mixing weight*. On each trial $t$, we denote the mixing weight associated with level $i$ by $\mu_t^i$. We note that the mixing weight $\mu_t^0$ is not necessary.

We now describe the creation of the action $\boldsymbol{x}_t$ on trial $t$. Note first that each level $i \in [\tau] \cup \{0\}$ has an associated action $\boldsymbol{w}_t^i$ which is defined as the output of $\text{QUERY}(\mathcal{D}_t^i)$, so is the action selected by the base algorithm for level $i$ on trial $t$. We call these actions the *base actions*. The action $\boldsymbol{x}_t$ is created by the following recursive process. For each level $i$ in order we construct an action $\boldsymbol{z}_t^i$ called the *propagating action*. This action is constructed by a convex combination of the preceding propagating action $\boldsymbol{z}_t^{i-1}$ and the base action $\boldsymbol{w}_t^i$. Specifically, we start by setting:

$$\boldsymbol{z}_t^0 \leftarrow \boldsymbol{w}_t^0$$

and then, for all levels $i \in [\tau]$, once $\boldsymbol{z}_t^{i-1}$ has been constructed we set:

$$\boldsymbol{z}_t^i \leftarrow \mu_t^i \boldsymbol{w}_t^i + (1 - \mu_t^i)\boldsymbol{z}_t^{i-1} \,.$$

Finally, we output:

$$\boldsymbol{x}_t \leftarrow \boldsymbol{z}_t^\tau \,.$$

We now turn to the update at the end of trial $t$. For all levels $i \in [\tau] \cup \{0\}$ we have the following two cases.

If $t \in 2^i \mathbb{N}$ then we set:

$$\mu_{t+1}^i \leftarrow 1/2 \quad ; \quad \mathcal{D}_{t+1}^i \leftarrow \text{INITIALISE}(2^i)$$

so that the mixing weight and instance of the base algorithm hosted by level $i$ are reset. Note that the parameter of the base algorithm is $2^i$. This is since it is reset every $2^i$ trials.

On the other hand, if $t \notin 2^i \mathbb{N}$ then we set:

$$\mu_{t+1}^i \leftarrow \psi(\mu_t^i, \ell_t(\boldsymbol{w}_t^i), \ell_t(\boldsymbol{z}_t^{i-1}), 2^i) \quad ; \quad \mathcal{D}_{t+1}^i \leftarrow \text{UPDATE}(\mathcal{D}_t^i, \ell_t) \,.$$

Note that the update of the mixing weight is based on the losses of $\boldsymbol{w}_t^i$ and $\boldsymbol{z}_t^{i-1}$. If $\boldsymbol{z}_t^{i-1}$ has a higher loss than $\boldsymbol{w}_t^i$, in that the base action performs better than the lower-level propagating action, then the mixing weight increases. This means that the weight of the base action, in the convex combination forming the propagating action of level $i$, increases. If $\boldsymbol{w}_t^i$ has a higher loss than $\boldsymbol{z}_t^{i-1}$ then the opposite happens.

Figure 1 illustrates RESET.

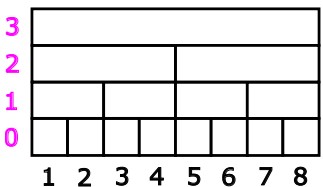 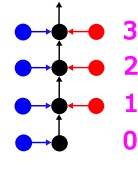

Figure 1: RESET with 8 trials.
Left: The generation of the base actions and mixing weights. Purple numbers denote levels and black numbers denote trials. Each segment (rectangle) in the figure runs an instance of the base algorithm (generating the base actions). The mixing weights for each level reset at the start of each segment. Mixing weight updates are dependent on the segment length.
Right: The computation of the action $\boldsymbol{x}_t$ on trial $t$. Purple numbers denote levels. Blue balls denote base actions, red balls denote mixing weights, and black balls denote propagating actions. The final black arrow is the output $\boldsymbol{x}_t$.

### 3.3 Comparison to Strongly Adaptive Online Learner

RESET has some similarities to the SAOL algorithm of [4]. Unlike RESET, SAOL is *strongly adaptive*, in that it bounds the static regret on any segment. Specifically, for any segment $\mathcal{I}$, SAOL achieves:

$$R(\mathcal{I}) \in \mathcal{O}\left(\ln(T)\sqrt{|\mathcal{I}|}\right) .$$

This, however, leads to a switching regret bound that is a factor $\mathcal{O}(\ln(T))$ off the optimal. This additional factor was improved to $\mathcal{O}(\sqrt{\ln(T)})$ by [8].

Like RESET, SAOL has $\tau + 1$ levels and utilises a base algorithm which constructs, for every trial $t$ and level $i$, a base action $\boldsymbol{w}_t^i$ in exactly the same way as RESET. It then generates, for each trial $t$, the final action $\boldsymbol{x}_t$ as a convex combination of the base actions. We note that in RESET, the action $\boldsymbol{x}_t$ is also a convex combination of the base actions, where for each level $i \in [\tau]$, the coefficient of $\boldsymbol{w}_t^i$ is equal to:

$$\mu_t^i \prod_{j=i+1}^{\tau} (1 - \mu_t^j) .$$

The crucial difference between SAOL and RESET is that, whilst SAOL updates each coefficient in the convex combination directly, the coefficients in RESET are updated by updating each mixing weight directly. It is due to this, and the particular way that the mixing weights are updated, that RESET attains, unlike SAOL, the optimal switching regret.

## 4 Analysis

Here we prove Theorem 2.2. We will show how to modify this proof in order to prove Theorem 2.4 at the end of this section. All lemmas stated in this section are proved in Appendix A.

Choose any segmentation $\mathcal{S}$. Let $\Phi := |\mathcal{S}|$ and for all $k \in [\Phi + 1]$ define $\sigma_k$ such that the $k$-th segment in $\mathcal{S}$ is $\langle \sigma_k, \sigma_{k+1} - 1 \rangle$. We define a comparator sequence $\langle \boldsymbol{\epsilon}_t \,|\, t \in [T] \rangle$ as follows. For all $k \in [\Phi]$ define the action:

$$\tilde{\boldsymbol{\epsilon}}_k := \operatorname{argmin}_{\boldsymbol{x}^* \in \mathcal{X}} \sum_{t=\sigma_k}^{\sigma_{k+1}-1} \ell_t(\boldsymbol{x}^*)$$

and then, for all $t \in \langle \sigma_k, \sigma_{k+1} - 1 \rangle$, define $\boldsymbol{\epsilon}_t := \tilde{\boldsymbol{\epsilon}}_k$. Note that:

$$R^\dagger(\mathcal{S}) = \sum_{t \in [T]} (\ell_t(\boldsymbol{x}_t) - \ell_t(\boldsymbol{\epsilon}_t)) . \tag{2}$$

In addition let $\alpha := 2\sqrt{\ln(2)}/(\sqrt{2} - 1)$. With these definitions in hand we now begin the analysis.

### 4.1 Hedge

The updates for our mixing weights follow the classic algorithm HEDGE [5]. In particular, for each level $i \in [\tau]$ we maintain an instance of HEDGE (which restarts every $2^i$ trials) with two *experts*. On each trial $t$, the weight of the first expert is $\mu_t^i$ and the weight of the second is $1 - \mu_t^i$. The loss of the first expert is $\ell_t(\boldsymbol{w}_t^i)$ and the loss of the second is $\ell_t(\boldsymbol{z}_t^{i-1})$. The purpose of the function $\psi$ is then to update the weights according to the HEDGE algorithm. The following lemma is a classic result about HEDGE.

**Lemma 4.1.** *Given trials $q, s \in [T]$ with $q \leq s$, and a sequence $\langle (a_t, b_t) \mid t \in \langle q, s \rangle \rangle$ such that for all $t \in \langle q, s \rangle$ we have $a_t, b_t \in [0, 1]$, and a sequence $\langle \rho_t \mid t \in \langle q, s \rangle \rangle$ defined recursively such that for all $t \in \langle q, s - 1 \rangle$ we have:*

$$\rho_q := 1/2 \quad ; \quad \rho_{t+1} := \psi(\rho_t, a_t, b_t, s - q + 1)$$

*then we have:*

$$\sum_{t=q}^{s} (\rho_t a_t + (1 - \rho_t) b_t) \leq \min \left\{ \sum_{t=q}^{s} a_t, \sum_{t=q}^{s} b_t \right\} + \sqrt{2 \ln(2)(s - q + 1)}.$$

### 4.2 The Segment Tree

In this subsection we define the *segment tree*, which is the geometrical structure that our analysis is based on. The segment tree is a full, balanced, binary tree $\mathcal{B}$ whose leaves are the elements of $[T]$ in order from left to right. Given any internal vertex $v \in \mathcal{B}$, let $\triangleleft(v)$ and $\triangleright(v)$ be its left and right child respectively. Given any vertex $v \in \mathcal{B}$, let $\blacktriangleleft(v)$ and $\blacktriangleright(v)$ be its left-most and right-most descendent respectively (noting that these are both elements of $[T]$). Let $r$ be the root of $\mathcal{B}$. Given a vertex $v \in \mathcal{B} \setminus \{r\}$, let $\uparrow(v)$ be the parent of $v$. Given a vertex $v \in \mathcal{B}$, let $h(v)$ be equal to the height of $v$ (that is, the height of the tree $\mathcal{B}$ minus the depth of $v$, so that leaves have height 0).

Each vertex $v \in \mathcal{B}$ represents the segment $\langle \blacktriangleleft(v), \blacktriangleright(v) \rangle$. i.e. Each vertex represents a segment in the left hand side of Figure 1 (when $t = 8$). We call a vertex $v \in \mathcal{B}$ *stationary* iff there exists $k \in [\Phi]$ with $\sigma_k \leq \blacktriangleleft(v)$ and $\blacktriangleright(v) < \sigma_{k+1}$. Let $\mathcal{H}$ be the set of all stationary vertices. We call a vertex $v \in \mathcal{B}$ *fundamental* iff both:

- $v \in \mathcal{H}$.
- $v = r$ or $\uparrow(v) \notin \mathcal{H}$.

Let $\mathcal{F}$ be the set of all fundamental vertices. We call a vertex $v \in \mathcal{B}$ *relevant* iff it is an ancestor of a fundamental vertex. Let $\mathcal{A}$ be the set of all relevant vertices. For all relevant vertices $v \in \mathcal{A}$ we define $\mathcal{Q}(v)$ to be the set of descendants of $v$ that are contained in $\mathcal{F}$.

We have, from the algorithm, the following lemma about the vertices of the segment tree:

**Lemma 4.2.** *Given any vertex $v \in \mathcal{B}$ we have:*

$$\blacktriangleright(v) - \blacktriangleleft(v) + 1 = 2^{h(v)}$$

*and:*

$$\mu_{\blacktriangleleft(v)}^{h(v)} = 1/2 \quad ; \quad \mathcal{D}_{\blacktriangleleft(v)}^{h(v)} = \text{INITIALISE}(2^{h(v)})$$

*and for all $t \in [\blacktriangleleft(v), \blacktriangleright(v) - 1]$ we have:*

$$\mu_{t+1}^{h(v)} = \psi\left(\mu_t^{h(v)}, \ell_t\left(\boldsymbol{w}_t^{h(v)}\right), \ell_t\left(\boldsymbol{z}_t^{h(v)-1}\right), 2^{h(v)}\right) \quad ; \quad \mathcal{D}_{t+1}^{h(v)} = \text{UPDATE}\left(\mathcal{D}_t^{h(v)}, \ell_t\right).$$

Note that this lemma shows that for all $v \in \mathcal{B}$ we run a single instance of both the base algorithm and HEDGE over the segment $\langle \blacktriangleleft(v), \blacktriangleright(v) \rangle$, as illustrated in Figure 1.

### 4.3 The Recursive Equations

We now derive the recursive equations that our analysis is based on. First note that for all $v \in \mathcal{F}$ there exists $\boldsymbol{u} \in \mathcal{X}$ such that $\boldsymbol{\epsilon}_t = \boldsymbol{u}$ for all $t \in \langle \blacktriangleleft(v), \blacktriangleright(v) \rangle$. Hence, Lemma 4.2 and Equation (1), and the fact that $\boldsymbol{w}_t^{h(v)}$ is the output of $\text{QUERY}(\mathcal{D}_t^{h(v)})$, lead to the following lemma.

**Lemma 4.3.** *For all $v \in \mathcal{F}$ we have:*

$$\sum_{t=\blacktriangleleft(v)}^{\blacktriangleright(v)} \ell_t\left(\boldsymbol{w}_t^{h(v)}\right) \leq \sum_{t=\blacktriangleleft(v)}^{\blacktriangleright(v)} \ell_t(\boldsymbol{\epsilon}_t) + \gamma\sqrt{2^{h(v)}} .$$

Note that, from the algorithm and the convexity of the loss functions, we have, for all $t \in [T]$ and all $v \in \mathcal{B}$ with $h(v) \neq 0$, that:

$$\ell_t\left(\boldsymbol{z}_t^{h(v)}\right) \leq \mu_t^{h(v)}\ell_t\left(\boldsymbol{w}_t^{h(v)}\right) + \left(1 - \mu_t^{h(v)}\right)\ell_t\left(\boldsymbol{z}_t^{h(v)-1}\right) .$$

So, by lemmas 4.2 and 4.1, we have the following lemma.

**Lemma 4.4.** *For all vertices $v \in \mathcal{B}$ with $h(v) \neq 0$ we have:*

$$\sum_{t=\blacktriangleleft(v)}^{\blacktriangleright(v)} \ell_t\left(\boldsymbol{z}_t^{h(v)}\right) \leq \min\left\{\sum_{t=\blacktriangleleft(v)}^{\blacktriangleright(v)} \ell_t\left(\boldsymbol{w}_t^{h(v)}\right) , \sum_{t=\blacktriangleleft(v)}^{\blacktriangleright(v)} \ell_t\left(\boldsymbol{z}_t^{h(v)-1}\right)\right\} + \sqrt{2\ln(2)2^{h(v)}} .$$

Noting that $\boldsymbol{z}_t^0 = \boldsymbol{w}_t^0$ for all $t \in [T]$, lemmas 4.3 and 4.4 immediately imply the following recursive equations. For all $v \in \mathcal{F}$ we have:

$$\sum_{t=\blacktriangleleft(v)}^{\blacktriangleright(v)} \ell_t\left(\boldsymbol{z}_t^{h(v)}\right) \leq \sum_{t=\blacktriangleleft(v)}^{\blacktriangleright(v)} \ell_t(\boldsymbol{\epsilon}_t) + \left(\gamma + \sqrt{2\ln(2)}\right)\sqrt{2^{h(v)}} \tag{3}$$

and for all $v \in \mathcal{A} \setminus \mathcal{F}$ we have:

$$\sum_{t=\blacktriangleleft(v)}^{\blacktriangleright(v)} \ell_t\left(\boldsymbol{z}_t^{h(v)}\right) \leq \sum_{t=\blacktriangleleft(v)}^{\blacktriangleright(v)} \ell_t\left(\boldsymbol{z}_t^{h(v)-1}\right) + \sqrt{2\ln(2)2^{h(v)}} . \tag{4}$$

## 4.4 Performing the Recursion

We now utilise equations (3) and (4) to perform the recursion. Specifically, we have the following inductive hypothesis for vertices in $\mathcal{A}$, which is proved by induction up the tree $\mathcal{B}$ from the vertices in $\mathcal{F}$ to the root. The reason it holds for vertices in $\mathcal{F}$ comes direct from Equation (3). For a vertex in $\mathcal{A} \setminus \mathcal{F}$, once the inductive hypothesis has been shown to hold for both its children, it is then shown to hold for the vertex itself by Equation (4). The inductive hypothesis is given in the following lemma.

**Lemma 4.5.** *For all $v \in \mathcal{A}$ we have:*

$$\sum_{t=\blacktriangleleft(v)}^{\blacktriangleright(v)} \ell_t(\boldsymbol{z}_t^{h(v)}) \leq \sum_{t=\blacktriangleleft(v)}^{\blacktriangleright(v)} \ell_t(\boldsymbol{\epsilon}_t) + \sum_{q \in \mathcal{Q}(v)} \left(\gamma + \sqrt{2\ln(2)} \sum_{k=0}^{h(v)-h(q)} \sqrt{2^{-k}}\right)\sqrt{2^{h(q)}} .$$

In particular, this inductive hypothesis holds for $v = r$. By Equation (2) this gives us the following.

**Lemma 4.6.** *We have:*

$$R^\dagger(\mathcal{S}) \leq (\gamma + \alpha)\sum_{q \in \mathcal{F}} \sqrt{2^{h(q)}} .$$

We now have a bound on the switching regret. It is, however, not yet written in terms of the segment lengths. To write it in terms of the segment lengths we partition $\mathcal{F}$ into a sequence of sets $\langle \mathcal{F}_k \mid k \in [\Phi] \rangle$ such that, for all $k \in [\Phi]$, we define $\mathcal{F}_k$ to be equal to the set of all $v \in \mathcal{F}$ such that $\sigma_k \leq \blacktriangleleft(v)$ and $\blacktriangleright(v) < \sigma_{k+1}$. We now have the following lemma.

**Lemma 4.7.** *For all $k \in [\Phi]$ we have:*

$$\sum_{v \in \mathcal{F}_k} \sqrt{2^{h(v)}} \leq c\sqrt{\sigma_{k+1} - \sigma_k} .$$

Combining lemmas 4.6 and 4.7 gives us:

$$R^\dagger(\mathcal{S}) \leq (\gamma + \alpha)\sum_{v \in \mathcal{F}} \sqrt{2^{h(v)}} = (\gamma + \alpha)\sum_{k \in [\Phi]}\sum_{v \in \mathcal{F}_k} \sqrt{2^{h(v)}} = c(\gamma + \alpha)\sum_{k \in [\Phi]} \sqrt{\sigma_{k+1} - \sigma_k}$$

as required. This completes the proof of Theorem 2.2.

## 4.5 Dynamic Regret Analysis

We now prove Theorem 2.4. Let the base algorithm be online gradient descent as in [11]. Take any segmentation $\mathcal{S}^*$ and any comparator sequence $\mathcal{E}$. Let $\Psi = |\mathcal{S}^*|$ and for all $j \in [\Psi]$ let $\mathcal{I}_j$ be the $j$-th segment in $\mathcal{S}^*$. For all $j \in [\Psi]$, note that we can choose a natural number:

$$N_j \leq 1 + P(\mathcal{E}, \mathcal{I}_j)$$

and a partition $\mathcal{S}'_j$ of $\mathcal{I}_j$ into $N_j$ segments such that for all $\mathcal{I} \in \mathcal{S}'_j$ we have:

$$P(\mathcal{E}, \mathcal{I}) \in \mathcal{O}(1) \tag{5}$$

Now define the segmentation:

$$\mathcal{S} := \bigcup_{j \in [\Psi]} \mathcal{S}'_j$$

Note that Equation (5) implies that for all $\mathcal{I} \in \mathcal{S}$ we have:

$$P(\mathcal{E}, \mathcal{I}) \in \mathcal{O}(1) \,. \tag{6}$$

We now modify the analysis of the switching regret as follows. In the analysis of the switching regret we defined a comparator sequence $\langle \boldsymbol{\epsilon}_t \,|\, t \in [T] \rangle$. In this analysis we instead define this comparator sequence as equal to $\mathcal{E}$. Using the segmentation $\mathcal{S}$, construct the segment tree and the sets $\mathcal{A}$ and $\mathcal{F}$ as in the analysis of the switching regret. Since our base algorithm is gradient descent we have, direct from [11], the following lemma.

**Lemma 4.8.** *Suppose we have trials $q, s \in [T]$ with $s \geq q$, and on each trial $t \in \langle s, q \rangle$ we do the following:*

1. *If $t = s$ then $\mathcal{D}_t \leftarrow \text{INITIALISE}(q - s + 1)$.*

2. *$\boldsymbol{w}_t \leftarrow \text{QUERY}(\mathcal{D}_t)$.*

3. *$\mathcal{D}_{t+1} \leftarrow \text{UPDATE}(\mathcal{D}_t, \ell_t)$.*

*Then we have:*

$$\sum_{t=q}^{s} (\ell_t(\boldsymbol{w}_t) - \ell_t(\boldsymbol{\epsilon}_t)) \in \mathcal{O}\left( (1 + P(\mathcal{E}, \langle q, s \rangle)) \sqrt{q - s + 1} \right) \,.$$

This lemma, along with Lemma 4.2 and Equation (6), gives us the following.

**Lemma 4.9.** *For all $v \in \mathcal{F}$ we have:*

$$\sum_{t=\blacktriangleleft(v)}^{\blacktriangleright(v)} \ell_t\left( \boldsymbol{w}_t^{h(v)} \right) \leq \sum_{t=\blacktriangleleft(v)}^{\blacktriangleright(v)} \ell_t(\boldsymbol{\epsilon}_t) + \mathcal{O}\left( \sqrt{2^{h(v)}} \right) \,.$$

This lemma is essentially identical to, and will be used instead of, Lemma 4.3. Following the rest of the analysis of the switching regret gives us:

$$R^*(\mathcal{E}) \in \mathcal{O}\left( \sum_{\mathcal{I} \in \mathcal{S}} \sqrt{|\mathcal{I}|} \right) = \mathcal{O}\left( \sum_{j \in [\Psi]} \sum_{\mathcal{I} \in \mathcal{S}'_j} \sqrt{|\mathcal{I}|} \right) \,.$$

Note that for all $j \in [\Psi]$ we have $|\mathcal{S}'_j| = N_j$ and hence:

$$\sum_{\mathcal{I} \in \mathcal{S}'_j} \sqrt{|\mathcal{I}|} \leq \sqrt{N_j \sum_{\mathcal{I} \in \mathcal{S}'_j} |\mathcal{I}|} = \sqrt{N_j |\mathcal{I}_j|} \leq \sqrt{(1 + P(\mathcal{E}, \mathcal{I}_j)) |\mathcal{I}_j|}$$

so we have proved Theorem 2.4.

## Acknowledgements

Research funded by the Defence Science and Technology Laboratory (Dstl) which is an executive agency of the UK Ministry of Defence providing world class expertise and delivering cutting-edge science and technology for the benefit of the nation and allies. The research supports the Autonomous Resilient Cyber Defence (ARCD) project within the Dstl Cyber Defence Enhancement programme.

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

# A  Proofs

Here we prove, in order, all the lemmas in the analysis.

## A.1  Lemma 4.1

Direct from [5] using only two experts, were, on each trial $t$, we have that:

- The loss of the first expert is $a_t$ and the loss of the second is $b_t$.
- The weight of the first expert is $\rho_t$ and the weight of the second is $1 - \rho_t$.

## A.2  Lemma 4.2

The equality:

$$\blacktriangleright(v) - \blacktriangleleft(v) + 1 = 2^{h(v)}$$

comes directly from the definition of $h(v)$. We also have that:

$$\blacktriangleleft(v) - 1 \in 2^{h(v)}\mathbb{N}$$

and for all $t \in [\blacktriangleleft(v), \blacktriangleright(v) - 1]$ we have:

$$t \notin 2^{h(v)}\mathbb{N}.$$

Hence, by the algorithm, the lemma holds.

## A.3 Lemma 4.3

Note first that, since $\boldsymbol{w}_t^{h(v)}$ is the output of $\text{QUERY}(\mathcal{D}_t^{h(v)})$, Lemma 4.2 and Equation (1) immediately give us:

$$\max_{\boldsymbol{x}^* \in \mathcal{X}} \sum_{t=\blacktriangleleft(v)}^{\blacktriangleright(v)} \left( \ell_t\left(\boldsymbol{w}_t^{h(v)}\right) - \ell_t(\boldsymbol{x}^*) \right) \leq \gamma\sqrt{2^{h(v)}} \,. \tag{7}$$

Since $v \in \mathcal{F}$ we have that $v \in \mathcal{H}$ so there exists $k \in [\Phi]$ with $\sigma_k \leq \blacktriangleleft(v)$ and $\blacktriangleright(v) < \sigma_{k+1}$. This implies that for all $t \in \langle\blacktriangleleft(v), \blacktriangleright(v)\rangle$ we have $t \in \langle\sigma_k, \sigma_{k+1} - 1\rangle$ so that $\boldsymbol{\epsilon}_t := \tilde{\boldsymbol{\epsilon}}_k$. Hence, we have:

$$\min_{\boldsymbol{x}^* \in \mathcal{X}} \sum_{t=\blacktriangleleft(v)}^{\blacktriangleright(v)} \ell_t(\boldsymbol{x}^*) \leq \sum_{t=\blacktriangleleft(v)}^{\blacktriangleright(v)} \ell_t(\tilde{\boldsymbol{\epsilon}}_k) = \sum_{t=\blacktriangleleft(v)}^{\blacktriangleright(v)} \ell_t(\boldsymbol{\epsilon}_t) \,.$$

Substituting into Equation (7) gives us the result.

## A.4 Lemma 4.4

Consider Lemma 4.1. In this lemma choose $q := \blacktriangleleft(v)$ and $s := \blacktriangleright(v)$. For all $t \in \langle q, s \rangle$ choose:

$$a_t := \ell_t\left(\boldsymbol{w}_t^{h(v)}\right) \quad ; \quad b_t := \ell_t\left(\boldsymbol{z}_t^{h(v)-1}\right) \,.$$

Then by Lemma 4.2 we have, for all $t \in [q, s]$, that:

$$\rho_t = \mu_t^{h(v)}$$

so that, by the algorithm and the convexity of the loss functions we have, for all $t \in \langle q, s \rangle$, that:

$$\ell_t\left(\boldsymbol{z}_t^{h(v)}\right) \leq \mu_t^{h(v)} \ell_t\left(\boldsymbol{w}_t^{h(v)}\right) + \left(1 - \mu_t^{h(v)}\right) \ell_t\left(\boldsymbol{z}_t^{h(v)-1}\right) = \rho_t a_t + (1 - \rho_t)b_t$$

and hence, by Lemma 4.1, we have:

$$\sum_{t=q}^{s} \ell_t\left(\boldsymbol{z}_t^{h(v)}\right) \leq \min\left\{\sum_{t=q}^{s} a_t \,,\, \sum_{t=q}^{s} b_t\right\} + \sqrt{2\ln(2)(s - q + 1)} \,.$$

The result then follows from the fact that, by Lemma 4.2, we have $s - q + 1 = 2^{h(v)}$.

## A.5 Lemma 4.5

For $k \in \mathbb{N} \cup \{0\}$ we define:

$$\lambda_k := \sum_{i=0}^{k} \sqrt{2^{-i}}$$

so our inductive hypothesis is that for all $v \in \mathcal{A}$ we have:

$$\sum_{t=\blacktriangleleft(v)}^{\blacktriangleright(v)} \ell_t\left(\boldsymbol{z}_t^{h(v)}\right) \leq \sum_{t=\blacktriangleleft(v)}^{\blacktriangleright(v)} \ell_t(\boldsymbol{\epsilon}_t) + \sum_{q \in \mathcal{Q}(v)} \left(\gamma + \lambda_{h(v)-h(q)}\sqrt{2\ln(2)}\right)\sqrt{2^{h(q)}} \,.$$

If $v \in \mathcal{F}$ then, by the definition of $\mathcal{F}$, we have $\mathcal{Q}(v) = \{v\}$ so the inductive hypothesis holds by Equation (3). Now suppose we have some $\tilde{v} \in \mathcal{A} \setminus \mathcal{F}$. Note that, by the definition of $\mathcal{F}$, we have $\lhd(\tilde{v}), \rhd(\tilde{v}) \in \mathcal{A}$ so all that is left to prove the inductive hypothesis is to prove that, if the inductive hypothesis holds for both $v = \lhd(\tilde{v})$ and $v = \rhd(\tilde{v})$, then it also holds for $v = \tilde{v}$. So suppose that the inductive hypothesis holds for $v = \lhd(\tilde{v})$ and $v = \rhd(\tilde{v})$. First note that:

$$\sum_{t=\blacktriangleleft(\tilde{v})}^{\blacktriangleright(\tilde{v})} \ell_t\left(\boldsymbol{z}_t^{h(\tilde{v})-1}\right) = \sum_{t=\blacktriangleleft(\lhd(\tilde{v}))}^{\blacktriangleright(\lhd(\tilde{v}))} \ell_t\left(\boldsymbol{z}_t^{h(\lhd(\tilde{v}))}\right) + \sum_{t=\blacktriangleleft(\rhd(\tilde{v}))}^{\blacktriangleright(\rhd(\tilde{v}))} \ell_t\left(\boldsymbol{z}_t^{h(\rhd(\tilde{v}))}\right) \tag{8}$$

and, by the definition of $\mathcal{F}$ (and a simple induction up the tree $\mathcal{B}$ from the vertices in $\mathcal{F}$), we have:

$$\sum_{q \in \mathcal{Q}(\tilde{v})} 2^{h(q)} = 2^{h(\tilde{v})}$$

so that:

$$\sum_{q \in \mathcal{Q}(\tilde{v})} \sqrt{2^{h(q)}} \sqrt{2^{h(q)-h(\tilde{v})}} = \sqrt{2^{-h(\tilde{v})}} \sum_{q \in \mathcal{Q}(\tilde{v})} 2^{h(q)} = 2^{h(\tilde{v})} \sqrt{2^{-h(\tilde{v})}} = \sqrt{2^{h(\tilde{v})}}. \qquad (9)$$

Equations (4), (8) and (9), imply:

$$\sum_{t=\blacktriangleleft(\tilde{v})}^{\blacktriangleright(\tilde{v})} \ell_t\left(\mathbf{z}_t^{h(\tilde{v})}\right) \leq \sum_{t=\blacktriangleleft(\vartriangleleft(\tilde{v}))}^{\blacktriangleright(\vartriangleleft(\tilde{v}))} \ell_t\left(\mathbf{z}_t^{h(\vartriangleleft(\tilde{v}))}\right) + \sum_{t=\blacktriangleleft(\vartriangleright(\tilde{v}))}^{\blacktriangleright(\vartriangleright(\tilde{v}))} \ell_t\left(\mathbf{z}_t^{h(\vartriangleright(\tilde{v}))}\right)$$
$$+ \sqrt{2\ln(2)} \sum_{q \in \mathcal{Q}(\tilde{v})} \sqrt{2^{h(q)}} \sqrt{2^{h(q)-h(\tilde{v})}}.$$

Applying the inductive hypothesis to the terms:

$$\sum_{t=\blacktriangleleft(\vartriangleleft(\tilde{v}))}^{\blacktriangleright(\vartriangleleft(\tilde{v}))} \ell_t\left(\mathbf{z}_t^{h(\vartriangleleft(\tilde{v}))}\right) \quad : \quad \sum_{t=\blacktriangleleft(\vartriangleright(\tilde{v}))}^{\blacktriangleright(\vartriangleright(\tilde{v}))} \ell_t\left(\mathbf{z}_t^{h(\vartriangleright(\tilde{v}))}\right)$$

and noting that $\mathcal{Q}(\tilde{v}) = \mathcal{Q}(\vartriangleleft(\tilde{v})) \cup \mathcal{Q}(\vartriangleright(\tilde{v}))$ and for all $q \in \mathcal{Q}(\tilde{v})$ we have:

$$\lambda_{h(\tilde{v})-h(q)} = \lambda_{h(\vartriangleleft(\tilde{v}))-h(q)} + \sqrt{2^{h(q)-h(\tilde{v})}} = \lambda_{h(\vartriangleright(\tilde{v}))-h(q)} + \sqrt{2^{h(q)-h(\tilde{v})}}$$

shows the inductive hypothesis holds for $v = \tilde{v}$. We have hence proved that the inductive hypothesis holds for all $v \in \mathcal{A}$.

## A.6 Lemma 4.6

For all $q \in \mathcal{Q}(r)$ we have:

$$\sum_{k=0}^{h(r)-h(q)} \sqrt{2^{-k}} \leq \sum_{k=0}^{\infty} \sqrt{2^{-k}} = \sqrt{2}/(\sqrt{2}-1) = \alpha/\sqrt{2\ln(2)}$$

so, since $r \in \mathcal{A}$ and $\mathcal{Q}(r) = \mathcal{F}$, Lemma 4.5 gives us:

$$\sum_{t=\blacktriangleleft(r)}^{\blacktriangleright(r)} \ell_t\left(\mathbf{z}_t^{h(r)}\right) \leq \sum_{t=\blacktriangleleft(r)}^{\blacktriangleright(r)} \ell_t(\boldsymbol{\epsilon}_t) + (\gamma + \alpha) \sum_{q \in \mathcal{F}} \sqrt{2^{h(q)}}.$$

Since $\blacktriangleleft(r) = 1$, $\blacktriangleright(r) = T$ and, by the algorithm, $\mathbf{z}_t^{h(r)} = \mathbf{z}_t^\tau = \mathbf{x}_t$ for all $t \in [T]$, we then have, by Equation (2), that:

$$R^\dagger(\mathcal{S}) \leq (\gamma + \alpha) \sum_{q \in \mathcal{F}} \sqrt{2^{h(q)}}$$

as required.

## A.7 Lemma 4.7

First let:

$$\xi := 1/(\sqrt{2}-1).$$

We take the inductive hypothesis that for all non-empty finite sets $\mathcal{Z} \subseteq \mathbb{N} \cup \{0\}$ we have:

$$\sum_{k \in \mathcal{Z}} \sqrt{2^k} \leq \xi \sqrt{\sum_{k \in \mathcal{Z}} 2^k} \qquad (10)$$

and we prove by induction on $|\mathcal{Z}|$. In the case that $|\mathcal{Z}| = 1$ we have $\mathcal{Z} = \{i\}$ for some $i \in \mathbb{N} \cup \{0\}$ and hence:

$$\sum_{k \in \mathcal{Z}} \sqrt{2^k} = \sqrt{2^i} < \xi \sqrt{2^i} = \xi \sqrt{\sum_{k \in \mathcal{Z}} 2^k}$$

so the inductive hypothesis holds for $|\mathcal{Z}| = 1$. Now suppose we have some $j \in \mathbb{N}$ and that the inductive hypothesis holds for $|\mathcal{Z}| = j$. We now show that it holds for $|\mathcal{Z}| = j + 1$ which will prove that the inductive hypothesis holds always. Specifically, let $i := \max \mathcal{Z}$, let $\mathcal{Z}' := \mathcal{Z} \setminus \{i\}$ and let $i' := \max \mathcal{Z}'$. Define:

$$y := 2^{-i} \sum_{k \in \mathcal{Z}'} 2^k .$$

Note that:

$$\sum_{k \in \mathcal{Z}'} 2^k \leq \sum_{k=0}^{i'} 2^k < 2^{i'+1} \leq 2^i$$

so that $y < 1$ and hence:

$$\sqrt{1 + y} - \sqrt{y} \geq \sqrt{1 + 1} - \sqrt{1} = \sqrt{2} - 1 = 1/\xi$$

since the term on the left is monotonic decreasing with $y$. This implies that:

$$\sqrt{\sum_{k \in \mathcal{Z}} 2^k} = \sqrt{2^i + \sum_{k \in \mathcal{Z}'} 2^k} = \sqrt{2^i}\sqrt{1 + y} \geq \sqrt{2^i}\left(\sqrt{y} + \frac{1}{\xi}\right) = \sqrt{\sum_{k \in \mathcal{Z}'} 2^k} + \frac{\sqrt{2^i}}{\xi}$$

so that, by the inductive hypothesis (applied to the set $\mathcal{Z}'$), we have:

$$\sqrt{\sum_{k \in \mathcal{Z}} 2^k} \geq \frac{1}{\xi} \sum_{k \in \mathcal{Z}'} \sqrt{2^k} + \frac{\sqrt{2^i}}{\xi} = \frac{1}{\xi} \sum_{k \in \mathcal{Z}} \sqrt{2^k}$$

so the inductive hypothesis holds for $|\mathcal{Z}| = j + 1$. We have hence proved that the inductive hypothesis holds always.

Now note that, by the definition of $\mathcal{F}$, we have that the set $\{\langle \blacktriangleleft(v), \blacktriangleright(v) \rangle \mid v \in \mathcal{F}_k\}$ is a partition of $\langle \sigma_k, \sigma_{k+1} - 1 \rangle$ so that, by Lemma 4.2, we have:

$$\sum_{v \in \mathcal{F}_k} 2^{h(v)} = \sigma_{k+1} - \sigma_k . \tag{11}$$

Assume, for contradiction, that there exists three distinct vertices $v, v', v'' \in \mathcal{F}_k$ with $h(v) = h(v') = h(v'')$. Without loss of generality assume that $v$ and $v''$ are the leftmost and rightmost of the three vertices respectively. Also without loss of generality assume that $v'$ is the right child of its parent $\uparrow(v')$. Then $v$ must either be equal to or lie to the left of the left child of $\uparrow(v')$ and hence:

$$\blacktriangleleft(\uparrow(v')) \geq \blacktriangleleft(v) \geq \sigma_k$$

and

$$\blacktriangleright(\uparrow(v')) = \blacktriangleright(v') < \sigma_{k+1}$$

so that $\uparrow(v') \in \mathcal{H}$. But this contradicts the fact that $v' \in \mathcal{F}$.

We have hence shown that for any $i \in [\tau] \cup \{0\}$ there are at most two distinct vertices $v, v' \in \mathcal{F}_k$ with $h(v) = h(v') = i$. This means that we can partition $\mathcal{F}_k$ into two disjoint sets $\mathcal{U}_k$ and $\mathcal{V}_k$ such that for all $i \in [\tau] \cup \{0\}$ there exists at most one element $v$ of $\mathcal{U}_k$ with $h(v) = i$ and at most one element $v'$ of $\mathcal{V}_k$ with $h(v') = i$. By Equation (10) we then have:

$$\sum_{v \in \mathcal{U}_k} \sqrt{2^{h(v)}} \leq \xi \sqrt{\sum_{v \in \mathcal{U}_k} 2^{h(v)}} \quad ; \quad \sum_{v \in \mathcal{V}_k} \sqrt{2^{h(v)}} \leq \xi \sqrt{\sum_{v \in \mathcal{V}_k} 2^{h(v)}}$$

so that:

$$\sum_{v \in \mathcal{F}_k} \sqrt{2^{h(v)}} \leq \xi \sqrt{\sum_{v \in \mathcal{U}_k} 2^{h(v)}} + \xi \sqrt{\sum_{v \in \mathcal{V}_k} 2^{h(v)}} .$$

Noting that for all $y, y' > 0$ we have $\sqrt{y} + \sqrt{y'} \leq \sqrt{2}\sqrt{y + y'}$ and $c = \xi\sqrt{2}$, the above inequality, along with Equation (11), gives us:

$$\sum_{v \in \mathcal{F}_k} \sqrt{2^{h(v)}} \leq c \sqrt{\sum_{v \in \mathcal{F}_k} 2^{h(v)}} = c\sqrt{\sigma_{k+1} - \sigma_k}$$

as required.

## A.8 Lemma 4.8

Direct from [11].

## A.9 Lemma 4.9

By lemmas 4.2 and 4.8 we immediately have that:

$$\sum_{t=\blacktriangleleft(v)}^{\blacktriangleright(v)} \left( \ell_t \left( \boldsymbol{w}_t^{h(v)} \right) - \ell_t(\boldsymbol{\epsilon}_t) \right) \in \mathcal{O} \left( (1 + P(\mathcal{E}, \langle \blacktriangleleft(v), \blacktriangleright(v) \rangle)) \sqrt{2^{h(v)}} \right). \tag{12}$$

Since $v \in \mathcal{F}$ we have that there exists $k \in [\Phi]$ such that $\blacktriangleleft(v) \geq \sigma_k$ and $\blacktriangleright(v) < \sigma_{k+1}$. Hence, by Equation (6), we have that:

$$P(\mathcal{E}, \langle \blacktriangleleft(v), \blacktriangleright(v) \rangle) \leq P(\mathcal{E}, \langle \sigma_k, \sigma_{k+1} - 1 \rangle) \in \mathcal{O}(1).$$

Substituting into Equation (12) gives us the result.

