# OpenReview forum: "Online Convex Optimisation: The Optimal Switching Regret for all Segmentations Simultaneously"
_NeurIPS.cc/2024/Conference — NeurIPS 2024 spotlight_

### Official Review · Reviewer_nGaq · 2024-06-21

**Soundness:** 2
**Presentation:** 1
**Contribution:** 2
**Rating:** 7
**Confidence:** 4

**Summary:**

This paper considers the problem of tracking regret (aka. switching cost). The previous start-of-the-art result has a $\ln T$ dependence on the final regret bound. This paper claims to remove the extra $\ln T$ (more specifically, an $\sqrt{\ln T}$ factor) and achieve the optimal switching regret.

**Strengths:**

The solution is clear and intuitive. I checked the whole proof and it seems to be correct. The combination way of $z_t^i = \mu_t^i w_t^i + (1 - \mu_t^i) z_t^{i-1}$ is impressive as it is different from the traditional weighted combination of $z_t^i = \sum_i \mu_t^i w_t^i$. This novel combination method has proven to play an important role in achieving the optimal switching regret. However, Neither the reason why this weighted combination method is able to achieve the desired optimal switching regret nor the comparison with the traditional combination method are explained in the paper. I suggest that the authors could revise the paper to add corresponding discussions.

**Weaknesses:**

I found the solution impressive. However, the current presentation of the paper is not satisfying enough. As stated in the above part, the reason why this weighted combination method is able to achieve the desired optimal switching regret and the comparison with the traditional combination method are explained in the paper, which are thought to be necessary from my point of view. I believe that the proofs are correct and the method is of importance. However, I am inclined that the current presentation does not meet the acceptance bar of NeurIPS and thus another round of polishment and review would be better.

**Questions:**

This paper can be further improved by considering the following minor issues:
1. The investigation and citations of the literature are insufficient. For example, in Line 18, when mirror descent is mentioned for the first time, there should be a citation, but not and only listing the related citations in the related work part.
2. Some writings are not consistent with standard practices. After the equation in Line 85, there should be a ',' followed by 'which' (not the capital 'Which' in the current version).
3. I believe that Theorem 2.1 exists in the literature as it is pretty straightforward and simple (by using the lower bound for static regret to attain a lower bound for the switching regret). However, there is no citation on this theorem. I suggest that the authors conduct a more careful check of the literature to refine this part.
4. The idea of 'multiple levels' in the proposed method is highly similar to the geometric covering proposed in [1], but without necessary discussions about the similarities or differences between them (I believe that the idea is the same). If they are similar, there should be a corresponding citation. The current presentation claims this idea to be one of the contributions of this work, which is misleading. Otherwise, if they are indeed different, I suggest that the authors could add a corresponding discussion in the next version.
5. The Section 'Analysis' is highly mathematical with little intuition. This section is more like a proof which should be put in the appendix. In the main paper, there should be more discussions about the proof sketch of the analysis, the reason why this paper uses such an algorithm, etc. I suggest that the authors could carefully polish the paper to enhance its readability.
6. The citations are somewhat inaccurate. For example, [3] (the index in the paper) is actually published at ICML 2015, but the current citation leads to arXiv. And [9] is actually published in NIPS 2018, but the current citation leads to arXiv.
7. The last part of dynamic regret is a little confusing. Do the authors mean that under the condition that the path length on each segment is bounded by a constant, a switching-regret-type dynamic regret can be achieved? If so, it does not make much sense because under such a strong condition, each segment is nearly stationary, and the dynamic regret degenerates to switching regret in this case.

References:
1. Strongly adaptive online learning, ICML 2015

**Limitations:**

The authors have adequately addressed the limitations and, if applicable, potential negative societal impact of their work.

---

> ### Author Rebuttal · Authors · 2024-08-06
>
> Thank you. We appreciate that you found our solution impressive and will improve the presentation of our paper, based on your feedback, as we detail below and in our comments to the other reviewers.
>
> “However, neither the reason why this weighted combination…”
> - We will add a sketch of the proof which will enable the reader to quickly see why it works. We will also add a comparison to previous methodologies.
>
> “and thus another round of polishment…”
> - We will make significant improvements to the presentation of our paper (moving the full proof to the appendix to give us space to do this) and are certain we can achieve the required NeurIPS standard for the camera-ready deadline.
>
> Questions:
>
> 1,2) Thank you, we will fix these things.
>
> 3) Theorems 2.1, 3.1 and 3.2 all exist in the literature and we state where they come from just before the theorem itself.
>
> 4) The segment tree structure is identical to the geometric covering structure. We didn’t give a citation since creating this binary tree structure over a chain is a standard mathematical technique that is used in many works. We in no way intended to suggest that the tree structure is novel to us and will add example citations. We agree that we should have cited [1] about the fact we have an instance of the base algorithm for each segment and that the tree reduces to different levels when the chain is the trial sequence.
>
> 5) Yes - as stated above we will include a clear and straightforward overview of how and why the algorithm works, deferring the analysis to the appendix.
>
> 6) We will fix.
>
> 7) Our dynamic regret definition is the same as in other works and our regret bound is a strict improvement over the current state of the art of $O(\sqrt{PT})$ where $P$ is path length (plus one). This can be seen since, given a comparator sequence with path length of $P$, we can partition the trial sequence into at most $P$ segments each with path length $O(1)$. If the length of the j-th segment is $L_j$ then our regret bound is $\mathcal{O}\left(\sum_{j\in[P]}\sqrt{L_j}\right)$. This is no greater than $\mathcal{O}\left(\sqrt{P\sum_{j\in[P]}L_j}\right)=\mathcal{O}\left(\sqrt{PT}\right)$.
> Note that we can massively improve on the $O(\sqrt{PT})$ result. For instance, suppose we have some $S$ much less than $T$, a path length of $P$ in the first $S$ trials and a path length of $1$ is the last $T-S$ trials. Then (by segmenting the first $S$ trials into $P$ segments of path length $O(1)$) our regret bound is at most $\mathcal{O}(\sqrt{PS}+\sqrt{T})$ - a massive improvement on $\mathcal{O}(\sqrt{PT})$. Our dynamic regret bound is equivalent to the following: given a segmentation such that the length of segment $j$ is $L_j$ and the path-length (plus 1) on segment $j$ is $P_j$ then our regret is $O\left(\sum_j\sqrt{P_jL_j}\right)$. We will add such discussion to the paper.

---

> ### Comment · Reviewer_nGaq · 2024-08-09
> **Thanks for the detailed feedback**
>
> Thanks to the authors for the detailed feedback. The authors have solved my major concerns. After carefully reading the paper once again, I am more convinced that this paper has made remarkable progress in the field of switching regret. Moreover, after reading the discussion of the authors with Reviewer #YiA9, I realize the significance of the results for the dynamic regret part. However, I think that Section 5 could be rewritten to make sure that Theorem 5.1 is not the final result for dynamic regret but an extension of the switching regret, which holds for **near-stationary** segmentations. And the authors could summarize another theorem (or maybe corollary) to illustrate that Theorem 5.1 suffices to achieve the desired $O(\sqrt{PT})$ dynamic regret, as explained in Question 7 above.
>
> I am also curious about whether it is possible to achieve problem dependent guarantees in switching regret using the method proposed in this work. For example, consider the well-known gradient variation in modern online learning: $V_T = \sum_{t=2}^T \sup_x \\|\nabla f_t(x) - \nabla f_{t-1}(x)\\|^2$ (please refer to [1,2,3] below). Gradient variation is essential due to its profound connections with stochastic/adversarial optimization, game theory, etc [4,5]. Is it possible to achieve its switching regret version of $\sum_k \sqrt{V_k}$, where $V_k$ denotes the gradient variation in the $k$-th segment? If not, what is the main challenge in achieving this?
>
> I raise my score to 7 and hope that the authors can make great efforts to improve the readability of this paper in the camera-ready version.
>
> **References:**
>
> [1] Online Optimization with Gradual Variations, COLT 2012
>
> [2] Universal Online Learning with Gradient Variations: A Multi-layer Online Ensemble Approach, NeurIPS 2023
>
> [3] Adaptivity and Non-stationarity: Problem-dependent Dynamic Regret for Online Convex Optimization, JMLR 2024
>
> [4] Between Stochastic and Adversarial Online Convex Optimization: Improved Regret Bounds via Smoothness, NeurIPS 2022
>
> [5] Fast Convergence of Regularized Learning in Games, NIPS 2015

---

> > ### Author Response · Authors · 2024-08-12
> >
> > We thank you for the score increase and your encouraging words about our result.
> >
> > We will give, in the camera ready, the final dynamic regret result: that, given any segmentation such that the length of segment $j$ is $L_j$ and the path length (plus one) on segment $j$ is $P_j$, then our regret is $O(\sum_j\sqrt{P_jL_j})$. This makes the essence of the result much clearer and clearly shows our improvement over the state of the art (which is achieved by considering the segmentation which contains the entire trial sequence as its single segment). We will also give the example we gave in the rebuttal to demonstrate that we can radically outperform the previous state of the art.
> >
> > With regards to the adaption to gradient variation… The base actions can made adaptive to gradient variation. However, the issue is with the aggregating step for constructing propagating actions, which (in its current form) introduces $O(\sqrt{L})$ factors (where $L$ is the length of a segment). We could use (instead of Hedge) for the aggregating step, an algorithm adaptive to gradient variation. Let $p_t$ and $q_t$ be the losses of the current base action and previous (i.e. lower level) propagating action on trial $t$ respectively. Then the goal of the aggregating step is to produce a normalised non-negative vector $(a,b)$, and the objective function on trial $t$ is $f_t(a,b)=ap_t+bq_t$. The derivative of the objective function is $(p_t,q_t)$. Hence, the aggregating step (for the node of the segment tree in question) will add a regret of $\mathcal{O}\left(\sqrt{\sum_{t\in S}((p_{t+1}-p_{t})^2+((q_{t+1}-q_t)^2)}\right)$ where $S$ is the segment associated with the node. This additional regret is based on the losses of the current base action and previous propagating action rather than the gradients of the loss functions. We will think on whether this can be further analysed.
> >
> > We will be spending significant time improving the readability of the paper and incorporating the other things that the reviewers have suggested.

---

### Official Review · Reviewer_YiA9 · 2024-06-28

**Soundness:** 2
**Presentation:** 1
**Contribution:** 4
**Rating:** 7
**Confidence:** 3

**Summary:**

This paper provides an algorithm which achieves a strongly-adaptive*
switching regret guarantee, removing the logarithmic penalty
typically associated with efficient algorithms achieving strongly-adaptive guarantees.

---
*Update after rebuttal: The results are only for switching regret, *not* strongly-adaptive switching regret guarantees

**Strengths:**

If the result is correct, it seems like it could be a breakthrough result,
removing a logarithmic penalty from strongly-adaptive guarantees that
seemed to be necessary.

**Weaknesses:**

It seems like the segment tree would still require $O(T)$ memory, would it not?
Whereas existing approaches achieve $O(\log(T))$ per-round memory and computation.
So it seems like the most interestin
So if the result is correct, perhaps the takeaway is that the log(T) penalty
is the cost of reducing the memory consumption of the algorithm.

The dynamic regret result doesn't seem particularly convincing since it only holds for
comparator sequences which do not move much, ie, for sequences that are essentially
just switching sequences. In fact it seems strange that any interesting
dynamic regret result could follow from a switching regret one, since the
optimal switching regret follows from the optimal $\sqrt{P_T T}$ dynamic regret
by throwing away information; if the comparator only changes at times
$\sigma_1,\ldots,\sigma_{S+1}$ then the path-length bound is $\sum_t \|u_t-u_{t-1}\|=\sum_i \|u_{\sigma_i}-u_{\sigma_{i+1}}\|\le S$ for $u\in\Delta_{d-1}$



The presentation is hard to follow. This work would also probably benefit from having diagrams
demonstrating the segment tree in the notation given here.

**Questions:**

- The approach seems very similar in spirit to the classic geometric covering intervals approach,
  which can also be seen as breaking up [1,T] according to a particular
  partition tree (see e.g. "Partition Tree Weighting" Veness et al. 2013).
  Could you explain what exactly the segment tree approach is doing to
  improve over the geometric covering intervals approach to avoid the log(T)
  penalty?
- Section 3.3 and Theorem 18 of Chen et al. (2021) ("Impossible Tuning Made Possible: A New Expert Algorithm and its Applications")
  seems to imply that a strongly-adaptive switching regret guarantee is not even possible in the first place.
  Could you comment on how this does not contradict the results presented here? Is this work leveraging some
  additional assumption?
- The result seems to forego all other types of adaptivity, and just accepts
  penalties of $\sqrt{b-a}$ on each interval $[b-a]$ instead of adapting to the
  losses of the experts. Do you think it is possible to get some form of
  adaptivity to the loss sequence as well, or is this a fundamental limitation of the approach?

**Limitations:**

The limitations are largely unaddressed. It's clear that some compromises are made to achieve this result,
but it's still very unclear to me what exactly they are, and these should have been laid out more clearly in the main text.

---

> ### Author Rebuttal · Authors · 2024-08-06
>
> Thank you, we appreciate that the importance of our work as a potential breakthrough was noted. We will improve the presentation of our paper, based on your feedback, as we detail below, and in our comments to the other reviewers.
>
> “If the result is correct, it seems like it could be a breakthrough…”
> - Thank you! :-) - we were very surprised ourselves when we did it. We can assure you that our proofs have been rigorously checked (as confirmed by Reviewer nGaq) and we are confident in our results.
>
> “It seems like the segment tree would still require…”
> - The segment tree is only used in the analysis and is not part of the algorithm, thus our memory requirement is only $O(\ln(T))$.
>
> “The dynamic regret doesn’t seem particularly convincing…”
> - We can certainly handle comparator sequences that move a lot and are a strict improvement over the previous state of the art of $O(\sqrt{PT})$ where $P$ is the path length (plus 1). To see why we outperform the the $O(\sqrt{PT})$ result note that, given the path length is $P$, we can split the trial sequence into at most P segments each with path length $O(1)$. If the j-th segment length is $L_j$ then our bound is $O\left(\sum_{j\in[P]} \sqrt{L_j}\right)$. This is never greater than $O\left(\sqrt{P\sum_{j\in[P]}L_j}\right)=O(\sqrt{PT})$.
> Note that we can massively improve on the $O(\sqrt{PT})$ result. For instance, suppose we have some $S$ much less than $T$, a path length of $P$ in the first $S$ trials and a path length of $1$ in the last $T-S$ trials. Then (by segmenting the first $S$ trials into $P$ segments of path length $O(1)$) our regret bound is at most $\mathcal{O}(\sqrt{PS}+\sqrt{T})$ - a massive improvement on $\mathcal{O}\sqrt{PT})$.
> We will add such discussion to the paper.
> - Our dynamic regret bound is equivalent to the following: given a segmentation such that the length of segment $j$ is $L_j$ and the path-length (plus 1) on segment $j$ is $P_j$ then our regret is $O\left(\sum_j\sqrt{P_jL_j}\right)$. We will add this to the paper.
>
> “In fact it seems strange that any interesting dynamic regret…”
> - The dynamic regret result doesn’t follow from the switching regret result but is proved in much the same way. Essentially, we take the result that gradient descent over a (known) segment of length L gives a dynamic regret of $O(\sqrt{L})$ when the path length is $O(1)$ - and substitute that into our analysis. We note that the optimal switching regret does not follow from the $O(\sqrt{PT})$ dynamic regret bound as this bound is not adaptive to heterogeneous segment lengths.
>
> “…probably benefit from having diagrams…”
> - Thank you for suggesting this - we will include diagrams in the camera ready paper - also to show the recursive generation of propagating actions.
>
> Questions…
>
> “The approach seems very similar in spirt…”
> - The segment tree structure itself is identical to the geometric covering intervals structure, a binary tree structure that appears in numerous works. What is novel about our algorithm is the computation over this tree. A use of e.g. specialist algorithms would maintain a weight for each of our base actions (constructed by mirror descent), play a convex combination (according to these weights) of the base actions, and then update the weights directly. However, we take a very different approach, recursively constructing “propagating actions” where each propagating action is defined by a convex combination of the previous (i.e. lower level) propagating action and the next (i.e. current level) base action. We then update the weights associated with these 2-action convex combinations. So we still in fact play a convex combination of base actions - it’s just that this convex combination is very different from other methodologies.
>
> “Section 3.3. of Chen et. al… seems to imply…”
> - This impossibility result is for the regret on any particular segment whereas our result is for the regret over the whole trial sequence. Note that the introduction of the seminal paper “strongly adaptive online learning” implies that the main reason for studying strongly adaptive regret (for any particular segment) was to obtain a bound on the whole trial sequence but with heterogenous segment lengths (as we do).
>
> “The result seems to forego all other types of adaptivity…”
> - We will certainly be able to incorporate other types of adaptivity. This is since our algorithm is based on two procedures: a base algorithm (e.g. mirror descent) and 2-expert Hedge - both of which can be made adaptive to various things. However, the final bound (whilst improving on our current bound) may not be able to be written so neatly as the current bound.
>
> Limitations…
>
> “It’s clear that some compromises are made…”
> - We solve the problem given at the start of Section 2.1 with no additional assumptions. We do of course need to be able to compute or approximate Bregman projections and sub-gradients if using mirror descent as our base algorithm, and have now clarified this limitation in the main text.

---

> > ### Comment · Reviewer_YiA9 · 2024-08-07
> >
> > Thank you for the detailed reply. After reading through it and the replies to the other reviewers, I am significantly more confident about the correctness of the results in this paper, and am convinced that this paper does indeed contain a breakthrough result, so I have decided to significantly raise my score.
> >
> > In addition to the clarifications given to my and others' reviews, I request the authors do more to distance this work from the notion of strongly-adaptive regret. Especially in the introduction, the discussion about "achieving $\sqrt{b-a}$ regret on all segments of length b-a simultaneously" implies strong adaptivity and is stronger than what is achieved here, because it implies independence of the segments: $R_{[a,b]}\le O(\sqrt{b-a})$ for any and all $a\le b$ simultaneously.
> > In contrast, while the results here do indeed adapt to each of the segments, the guarantee is for the sum of these regrets, $R_{[1,T]}\le \sum_i R_{[a_i,b_i]}\le \sum_i \sqrt{b_i-a_i}$, not for each segment independently.
> >
> > From the responses, I do believe that presentation is a *major* concern for the draft submitted. I think making the revisions suggested by each of the reviewers would massively improve the clarity of the paper. Many of the explanations given to the reviewers were convincing and should have been more clear in the main text; I would suggest moving some of the technical details to the appendix to make space to better address the key discussions. However, I do not think this point is grounds for rejection, and will point out that by the camera-ready deadline the authors will have had nearly half a year from the submission deadline to polish the presentation. Given the significance of the result, I think it should be presented where it will can have the highest impact, and am willing to give the authors the benefit of the doubt that the presentation can be raised to an acceptable standard for the venue before the camera-ready deadline.

---

> > > ### Author Response · Authors · 2024-08-08
> > >
> > > Thank you for your appreciative words about our result :-). We understand the difficulty to distinguish between our results and a fully strongly-adaptive one in the current manuscript, and we will clarify this explicitly for the camera-ready version. We can assure you and the other reviewers that we will bring the paper’s presentation fully up to standard, incorporating all the aspects raised in each of your reviews.

---

### Official Review · Reviewer_KotF · 2024-07-09

**Soundness:** 3
**Presentation:** 1
**Contribution:** 3
**Rating:** 6
**Confidence:** 3

**Summary:**

This paper studies online convex optimisation under a non-stationary environment. The authors focus on the switching regret, which evaluates the regret on every possible segment of the trials. This paper demonstrates that the additional logarithmic factor $O(\log T)$ shown in previous results can be improved, and the authors obtain an algorithm with an optimal guarantee.

**Strengths:**

This paper studies online convex optimisation under more challenging scenarios, and focuses on an important theoretical question: whether the additional logarithmic factor can be improved.

**Weaknesses:**

- As a theoretical paper that claims to make fundamental improvements, it would be beneficial for the descriptions of theorems to be more precise, rigorous, and comprehensive. For example, I would expect the authors to clarify under which assumptions ($\mathcal{X}$, loss functions) the theorems hold.
- The terminologies used in this paper are ambiguous. The term "switching regret" or "tracking regret" might often be used in a prediction with experts' advice setting, which could be treated as a degenerated version of dynamic regret, a measure considered widely for general convex optimization. The authors do not introduce the term "adaptive regret" [1, 2], which seems to be the same optimization objective in this paper, to my understanding. In [2], it is demonstrated that adaptive regret is stronger than tracking regret (or switching regret).
- There lacks a clear statement of the algorithm. It might improve the readability if the authors could explain the main idea and the key technique straightforwardly about how to achieve the optimal results.
- It would be better if the authors could provide more discussions about related works. See details in 'Questions'.

1. E. Hazan and C. Seshadhri. Efﬁcient learning algorithms for changing environments.
2. A. Daniely, A. Gonen, and S. Shalev-Shwartz. Strongly adaptive online learning.

**Questions:**

- Does Theorem 4.1 hold for any bounded convex set and Lipschitz convex loss functions? What is the meaning of $\gamma$ in Theorem 4.1?
- In Theorem 4 of [3], it is shown that the adaptive regret guarantee can recover the dynamic regret guarantee. Could the author explain the difference between the results achieved in Section 5 and those?
- What is the relationship between the proposed segment tree and the geometric covering [2]?
- At Line 144, it seems that new algorithms are initialized at some specific trials. The analysis of such an algorithm can be reduced to analyzing the sleeping expert problem [3], which usually requires the aggregating algorithm to have specific properties for minimizing general convex functions. Does the aggregating algorithm used in this paper have such properties?
- At Line 189, this paper analyzes the aggregating step in a black-box manner by using Theorem 3.2. The aggregating algorithm might be sensitive to the decision point at time $q$ in order to provide guarantees on any interval starting at time $q$. How does the proof avoid the analysis of this issue?

3. H. Luo and R. E. Schapire. Achieving all with no parameters: Adanormalhedge.

---

> ### Author Rebuttal · Authors · 2024-08-06
>
> Thank you for your review. We will improve the presentation of our paper based on your feedback, as we detail below, and in our comments to the other reviewers.
>
> “it would be beneficial for the descriptions of theorems to be more precise...”
> - All the conditions on $\mathcal{X}$ and the loss functions are given in the problem description (lines 69-71). The conditions are that $\mathcal{X}$ is a bounded convex subset of a Euclidean space and that the loss functions are convex with bounded gradients. Algorithmically though (i.e. to utilise mirror descent), we need to be able to compute Bregman projections into the set $\mathcal{X}$ (the euclidean projection will do to get the general bound) and compute sub-gradients (however, we only actually need to approximate these quantities to some degree). We will edit the theorem statements such that all requirements are contained inside them.
>
> “The terminologies used in this paper are ambiguous.”
> - Thank you. We will clarify the relationships between these different forms of regret in the camera-ready paper as follows.
> The difference between switching regret and dynamic regret is that, whilst dynamic regret considers a continually changing comparator sequence, switching regret considers a comparator sequence that changes at at most M trials. We chose to focus on switching regret for two reasons: (a) we can use any algorithm as our base algorithm (not just gradient descent) and (b) the result seems more profound from a theoretical perspective. However, when using gradient descent as our base algorithm our dynamic regret bounds imply our switching regret bounds so are stronger.
> Adaptive (and strongly adaptive - which takes into account the segment length) regret is when one bounds the regret on any possible segment - which is different from switching regret which bounds the cumulative regret (over all segments) for any segmentation. We do talk about strongly adaptive regret in lines 53-54. Like strongly-adaptive algorithms we adapt to heterogenous segment lengths (but unlike such algorithms we get rid of the $O(\ln(T))$ factor).
>
> "In [2], it is demonstrated that adaptive regret..."
> - The comparison to switching regret (a.k.a. tracking regret) in [2] is to the FixedShare bound which is not adaptive to heterogenous segment lengths (so is often very far from optimal). Our algorithm, however, is adaptive to heterogenous segment lengths.
>
> “It might improve the readability if the authors could explain the main idea and the key technique straightforwardly”
> We agree that the paper will benefit from a straightforward verbal explanation of our algorithm and will add this to the paper. We will also include a straightforward summary of the proof, as the proof is needed to explain why the algorithm works.
>
> “provide more discussions about related works”
> - Agreed, thank you. We will prepare an extended related works section for the final paper.
>
> Questions…
>
> “Does Theorem 4.1 hold for any bounded  convex set and Lipschitz convex loss functions?
> - Indeed!
>
> “What is the meaning of $\gamma$?”
> - We can utilise any base algorithm (e.g. mirror descent) with O(\sqrt{T}) static regret bound. $\gamma$ is the constant factor under that O. We will clarify this. For mirror descent we define $\gamma$ in Line 116.
>
> “Could the author explain the difference between the results achieved in Section 5 and those in Theorem 4 of [3]”
> - Their theorem contains the O(ln(T)) factor (which we remove) and is not adaptive to varying rates of change in the comparator sequence (which we are). However, we note that (unlike our switching regret bounds) our dynamic regret bounds hold only when using gradient descent as the base algorithm (which does not obtain the O(ln(N)) factor for experts). If the comparator sequence $\boldsymbol{u}$ in their theorem changes arbitrarily at M trials then the problem is switching regret - where we dramatically outperform by losing the O(ln(T)) factor and by being adaptive to segment lengths.
>
> “What is the relationship between the proposed segment tree and the geometric covering [2]”
> - There is no difference - the segment tree structure itself is used in many works. What is novel about our work is how the computations are performed over the segment tree.
>
> “At Line 144, it seems that new algorithms are initialised…”
> - The mechanics of our algorithm are very different from using a specialist (i.e. sleeping expert) algorithm to aggregate the base actions (defined on Line 139 and constructed via mirror descent) - which would not remove the O(ln(T)) factor. Utilising a specialist algorithm would (implicitly) require a weight for each segment in the segment tree (an object only used in our analysis) which would reduce to maintaining a weight for each base action. The final action would then use these weights to construct a convex combination of the base actions. The weights would then each be updated directly. However, as stated above, this would not remove the O(ln(T)) factor. Our algorithm, however, recursively constructs what we call ``propagating actions’’ where each such action is formed by a linear combination of the respective base action and the previous (i.e. lower level) propagating action. It is the weights involved in these two-action linear combinations that we update. This novel methodology is what allows us to remove the O(ln(T)) factor. Our method does not run into issues when going from linear functions to general convex functions.
>
> “At Line 189, this paper analyses the aggregating step…”
> - We are unsure what you mean here - could you rephrase please? This aggregating step uses a weight to merge the current propagating action and the current base action into the next (higher-level) propagating action. The weight is updated by the losses of both the current propagating action and the current base action. We are not trying to bound the regret on any particular segment/interval but rather the cumulative regret over the whole trial sequence.

---

> > ### Comment · Reviewer_KotF · 2024-08-08
> >
> > I thank the authors for their responses. I will restate the last question in my review. In Line 145, it appears that the base algorithm's decisions are reinitialized after certain rounds. Aggregating decisions from newly initialized experts usually requires the aggregating algorithm to provide stronger theoretical guarantees, such as a second-order bound [1]. I am confused about how the algorithm described after Line 128 could provide stronger theoretical guarantees than the classical Hedge algorithm, particularly in its capacity to accommodate newly initialized experts.
> >
> > The author's rebuttal and responses to other reviewers have resolved some of my concerns and clarified the problem addressed by this paper. I would appreciate it if the author could answer to the question I raised above.
> >
> > 1. Impossible Tuning Made Possible: A New Expert Algorithm and Its Applications, COLT 21.

---

> > > ### Author Response · Authors · 2024-08-08
> > >
> > > We are sorry but we are still confused as to exactly what you mean when you say “particularly in its capacity to accommodate newly initialized experts”. We do indeed only require the standard bound of Hedge for our analysis, and each time we utilise an instance of Hedge it is for the standard Hedge problem with two experts (the experts are action-valued rather than loss-valued but this modification does not damage the Hedge bound due to the convexity of the loss functions). Each instance of Hedge we use is run over a segment corresponding to a node in the segment tree. On each trial in that segment we have (for the specific instance of Hedge) a prediction issued by each of the two experts.
> > >
> > > We describe in detail the mechanics of the aggregating step, and how we utilise the Hedge bound, here. To analyse the aggregating step we need to look at it from the segment tree perspective. Each node of the segment tree corresponds to a segment of the trial sequence, and the segments associated with its two children partition its associated segment. We will use the words “segment” and “node” interchangeably in this description. Let’s say that a node is “active” if the current trial is within its corresponding segment. The active nodes correspond to the levels in the algorithm (the lowest level being the leaf). Whilst active, each node/segment contains a "base action" which changes from trial to trial. The base action is formed from mirror descent - which is started at the beginning of the segment (this corresponds to the reinitialisation in the algorithm) and ends at the end of the segment (which is the start of the next segment at that level of the tree and is hence the next reinitialisation). Each active node/segment also contains a "propagating action", which changes from trial to trial and is constructed (recursively via lower-level propagating actions) from the base actions of its descendants. Each active node/segment is associated with two experts (which issue actions as predictions). The first expert is the node/segment’s current base action. The second expert is the current propagating action of the child node that is currently active. Hence, whilst the node is active, there are always exactly two experts. We use Hedge (starting at the beginning of the segment and ending at the end of the segment) to form a convex combination of these two experts. This convex combination is the propagating action for the node/segment currently under discussion. So the (prediction sequence of the) second expert is formed from the concatenation (of the sequences of propagating actions) of the two child segments. This means that the cumulative loss of the second expert is the “cumulative loss of the propagating actions of the left child” plus the “cumulative loss of the propagating actions of the right child”. This can then be plugged into the Hedge bound to give a bound on the cumulative loss of the propagating actions of the node currently under discussion (with respect to those of its children). In our analysis we do this on all “relevant nodes” (defined on Line 175) that are not “fundamental nodes” (defined on Line 172). For fundamental nodes we also utilise the Hedge bound - but for the first expert (i.e. the base action) instead.  Via this methodology we obtain our recursive formula (Equations 5 and 6).
> > >
> > > We hope that this has answered your question. We will be adding, to the paper, a full discussion of how and why the algorithm works and will include a complete description of the aggregating step. We would again like to thank you for your comments on unclear parts of our presentation  and we are happy to provide further clarifications as desired.

---

> ### Comment · Reviewer_KotF · 2024-08-09
>
> Thank you for the authors' detailed response. I now have a clearer understanding of how to aggregate base algorithms, which has increased my confidence about this paper. After reevaluating the results of this paper, I realize that the results are indeed interesting to the community. The paper studies another metric that interpolates between adaptive regret and dynamic regret, which prompted me to raise my score. I hope the author can improve the writing in future versions.
>
> As noted by other reviewers, I believe the writing in this paper requires substantial revision to enhance its readability and impact. Given the close relevance of the paper's results and techniques to dynamic regret and adaptive regret, it is crucial to include a dedicated section discussing the connections and distinctions between these measures. It would be beneficial to emphasize early on that the aggregation method used here differs from that in SAOL [1]. To the best of my knowledge, [2] is the first paper to *sequentially* aggregate base algorithms to achieve adaptive regret and dynamic regret. It may be worthwhile to consider comparing the proposed methods with the approach in [2]. In Section 3, providing a formal algorithmic description would improve readability, at least for me. In the analysis sections, I suggest summarizing the analysis into specific lemmas, explaining the significance of each lemma, and how they collectively lead to the final conclusions. More explanation should be provided regarding the analysis of $\mathcal{Q}(v)$; in its current form, I found it challenging to catch the underlying idea.
>
> Additionally, I have another question concerning the technique. Recent work has explored the dynamic regret of exp-concave functions [3], which include important functions in machine learning, such as logistic loss and log loss. I am interested in whether the techniques presented in this paper could be adapted to derive the switching regret bound for exp-concave functions as well.
>
> 1. Strongly adaptive online learning, ICML’15.
> 2. Parameter-free, dynamic, and strongly-adaptive online learning, ICML’20.
> 3. Optimal dynamic regret in exp-concave online learning, COLT’21.

---

> > ### Author Response · Authors · 2024-08-13
> >
> > Thank you - we shall be incorporating all your suggestions into the camera-ready. As to the exp-concave question - let us read [3], think about it, and get back to you in the post-discussion period.

---

### Official Review · Reviewer_HMtq · 2024-07-12

**Soundness:** 3
**Presentation:** 3
**Contribution:** 3
**Rating:** 6
**Confidence:** 4

**Summary:**

The paper considers the problem of switching and dynamic regret in online convex optimization (OCO). Given any segmentation of $[T]$, the switching regret is equal to the sum of static regrets on each segment. For the switching regret, the best-known bound obtained by prior works was $\mathcal{O}(\sum_{k}\sqrt{\Lambda_{k} \ln T})$, where $\Lambda_{k}$ is the length of the $k$-th segment. However, the lower bound is $\Omega(\sum_{k} \sqrt{\Lambda_{k}})$, which follows by independently applying the lower bound on the static regret of OCO with convex functions, on each segment. The current paper removes the extra $\sqrt{\ln T}$ factor by proposing an efficient algorithm RESET (Recursion over Segment Tree), which has a running time $\mathcal{O}(\ln T)$ per iteration (disregarding the cost associated with treating the feasible set).

For the dynamic regret, it is known from Zhang et al. that a dynamic regret bound of $\mathcal{O}(\sqrt{\mathcal{P}T})$, where $\mathcal{P}$ denotes the path-length of the comparator sequence, is possible. However, the bound is not sensitive to variation in the path-length. The current work improves upon this by showing that RESET achieves $\mathcal{O}(\sum_{k} \sqrt{\Lambda_k})$ dynamic regret for any segmentation that has $\mathcal{O}(1)$ path length on each segment. When the segments have varying lengths, this bound is significantly better.

**Strengths:**

The considered problem is theoretically interesting and the presented algorithm is quite simple with a clean analysis. The writing is to the point. The idea of having a segment tree is natural since segment trees are known to handle interval updates, e.g. sum-query updates over intervals in an array efficiently. The subroutines used by RESET are Mirror Descent and Hedge. The algorithm can be viewed as updates over the (level, time) space, i.e. at each time $t$ we update the parameters $w_{t} ^ {i}, \mu_{t} ^ {i}, z_{t} ^ {i}$ by Mirror Descent, Hedge, a convex combination (with parameter $\mu_{t} ^ i$) of $z_{t}^{i - 1}$ and $w_{t} ^ {i}$. However, the analysis proceeds by translating these updates to a recursion defined over the nodes of a segment tree; the recursion is obtained by the updates and the regret guarantees of Mirror Descent and Hedge. The dynamic regret guarantee follows straightforwardly from the analysis of switching regret.

**Weaknesses:**

I do not see any immediate weakness regarding the contribution of the work. Although the presentation of the paper is good, there is a scope for improvement. I suggest the authors use appropriate punctuation marks to end/continue equations and sentences.

Minor Typo: Definition of Bregman Divergence in Section 3.1.

**Questions:**

Lines 183--184 are not very clear to me. If I imagine a full, balanced tree with 15 nodes ($T = 8$) with all levels except the leaf nodes being filled, I do not see how 183 holds when $v$ is the parent of the node with the number $7$. Is that a typo?

**Limitations:**

Yes, the authors have adequately addressed this.

---

> ### Author Rebuttal · Authors · 2024-08-06
>
> We appreciate that the theoretical importance of our results was acknowledged and thank you for your review.
>
> “I suggest the authors use appropriate punctuation”
> - We will ensure appropriate punctuation is used throughout the final camera-ready version of our manuscript.
>
> “Minor typo…”
> - We have fixed the definition of Bregman divergence to include prime on the u in the gradient.
>
> “Lines 183-184 are not very clear to me”
> We have clarified lines 183-184 in our paper as follows.
> - Note that $\blacktriangleleft(v)$ and $\blacktriangleright(v)$ are the leftmost and rightmost descendants of $v$ respectively, and $h(v)$ is the height of $v$ (where the height of a leaf is $0$). Also note that $j\mathbb{N}$ is the set of all positive multiples of $j$.
> For your example we have $v$ as the parent of the 7th leaf (node 7). Since $v$ is the parent of a leaf we have $h(v)=1$ so that $2^{h(v)}=2$ and hence $2^{h(v)}\mathbb{N}$ is the set of even numbers. The leftmost and rightmost descendants of $v$ are 7 and 8 respectively. So Line 183 states that 7-1=6 is even and Line 184 states that all $t\in[7,8-1]$ (so just $t=7$) are odd.

---

> > ### Comment · Reviewer_HMtq · 2024-08-10
> >
> > Thanks for your reply. I went through the author's responses to the other reviews and do not have any further questions. I appreciate the result, however, as other reviewers suggested, please consider polishing the paper in the subsequent revisions.

---

> > > ### Author Response · Authors · 2024-08-13
> > >
> > > Thank you - we will polish the paper and incorporate the suggestions of all the reviewers.

---

### Decision · Program_Chairs · 2024-09-25

**Decision:**

Accept (spotlight)

**Comment:**

This paper studies the problem of optimizing switching regret in online convex optimization and provides novel and interesting results. The proposed algorithm achieves optimal switching regret across all possible segmentations simultaneously with $\log T$ space and time complexity. The reviewers unanimously agreed that the approach is non-trivial and that the results are important for the community. Overall, the contribution is significant, but the reviewers have additional suggestions for further refining the presentation. The authors are requested to take these comments into consideration.